# Tensile Properties and Damping Capacity of Cold-Rolled Fe-20Mn-12Cr-3Ni-3Si Damping Alloy

**DOI:** 10.3390/ma14205975

**Published:** 2021-10-11

**Authors:** Jae-Hwan Kim, Jong-Min Jung, Hyunbo Shim

**Affiliations:** 1Fusion Energy Research and Development Directorate, National Institutes for Quantum and Radiological Science and Technology, QST, Aomori 039-3212, Japan; kim.jaehwan@qst.go.jp; 2Department of Industrial Facility Automation, Ulsan Campus of Korea Polytechnic, Ulsan 44482, Korea; jmjung@kopo.ac.kr; 3Heavy Plate R&D Team, Hyundai Steel, Dangjin 31719, Korea

**Keywords:** damping, martensite, cold rolling, tensile properties

## Abstract

The tensile properties and damping capacity of cold-rolled Fe–20Mn–12Cr–3Ni–3Si alloys were investigated. The martensitic transformation was identified, including surface relief with a specific orientation and partial intersection. Besides, as the cold rolling degree increased, the volume fraction of ε-martensite increased, whereas α’-martensite started to form at the cold rolling degree of 15% and slightly increased to 6% at the maximum cold rolling degree. This difference may be caused by high austenite stability by adding alloying elements (Mn and Ni). As the cold rolling degree increased, the tensile strength linearly increased, and the elongation decreased due to the fractional increment in the volume of martensite. However, the damping capacity increased until a 30% cold rolling degree was approached, and then decreased. The irregular tendency of the damping capacity was confirmed, depicting that it increased to a specific degree and then decreased as the tensile strength and elongation increased. Concerning the relationship between the tensile properties and the damping capacity, the damping capacity increased and culminated, and then decreased as the tensile properties and elongation increased. The damping capacity in the high-strength area tended to decrease because it is difficult to dissipate vibration energy into thermal energy in alloys with high strength. In the low-strength area, on the other hand, the damping capacity increased as the strength increased since the increased volume fraction of ε-martensite is attributed to the increase in the damping source.

## 1. Introduction

In the rapid development process of modern industrial society, vibration and noise has caused various human and material losses and is a social problem in improving the working environment, high value-added products, the pursuit of a comfortable living environment, and the stability and long life of equipment or devices. Great interest has been paid to the necessity and importance of reducing and noise reduction. Therefore, various studies have been conducted to prevent vibration and noise.

Damping technology is largely classified into system, structure, and material aspects. In the system damping method, a damper with a large damping coefficient is externally installed on a vibrating body that produces noise. In the structural damping method, including materials, the workability and weldability between the base and dissimilar materials are identified as problems, so it is very limited in its use. Therefore, research activities have been focused on the material aspect to maximize the noise and vibration suppression effect (damping ability) by applying metallic materials [1,2] and composite materials [3] with large internal friction coefficients to the vibration source. Furthermore, these have excellent working properties and adhesiveness, and related research activities have been actively conducted [4,5].

Generally, since steel exhibits a stress-induced martensitic transformation behavior in which austenite is transformed into martensite at room temperature [6], it is processed into martensite and then reverse-transformed to make ultrafine austenite or austenite/martensite dual-phases. Research [4] has been conducted to expand the scope of use by making dual-phase structural steel materials with the excellent combination of strength and elongation.

Furthermore, the martensitic transformation dramatically affects the vibration damping ability [7] when the γ-austenite transforms into martensite by processing. Consequently, when the strength of a material is high, it is difficult to dissipate its vibrational energy as heat energy, and its damping ability is low [5,8]. However, this relationship varies depending on the material’s damping mechanism. It is indispensable to investigate the relationship between mechanical properties and the damping capacity according to the deformation-induced martensitic transformation, in order to expand the range of use of these steel materials and secure stability in their use. In Fe–Mn-based alloys, γ-austenite is generally formed at room temperature, and it is related to controlling the proportion of the α’/ε-martensite formed through cold working and heat treatment for strength improvement [9,10,11]. However, for the damping capacity, the control of the ε-martensite fraction is dominant [12,13].

Herein, a damping alloy (Fe-20Mn-12Cr-3Ni-3Si) was designed and synthesized to develop practical Fe–Mn-based alloys with excellent properties. Then, the mechanical properties and damping capacity of this alloy according to the cold rolling degree were investigated. Furthermore, the relationship between the tensile properties and damping capacity was clarified.

## 2. Materials and Methods

Specimens were fabricated via melting in a vacuum melting furnace to produce an ingot. After that, it was hot-rolled at 1200 °C to produce 3 mm, 4 mm, and 10 mm-thick plates. After the solution treatment, including heat treatment at 1050 °C for 1 h and water cooling, the prepared plates were subjected to surface pickling (surface pickling conditions) and cold rolled to a final thickness of 2 mm with different cold rolling degrees of 13, 29, 49, and 69%. The chemical composition of the test specimen is shown in Table 1.

The microstructures of each specimen were observed using an optical microscope (OM) after surface etching with an etching solution of 5% hydrochloric acid (HCl), 5% nitric acid (HNO_3_), and 90% methyl alcohol (CH_3_OH). Furthermore, scanning (SEM) and transmission (TEM) electron microscopy (JEM-2010. 200 kV, JEOL, Tokyo, Japan) were used to observe the detailed microstructure and pattern for qualitative analyses. For the TEM, the specimen was <100 μm-thick and was jet-polished using a solution of 90% acetic acid (CH_3_COOH) and 10% perchloric acid (HClO_4_).

For quantitative analyses of each phase formed during cold rolling of the solution-treated specimens, X-ray diffraction tests were performed in the range of 10° to 80° with Cu Kα radiation. In addition, the phase volume fractions were evaluated as the sum of specific peak-integrated intensities of each phase [14]. However, for the α’-martensitic phase, it was difficult to measure the phase fraction with the peak cross-sectional area because of the minute area, so the phase-mapping method of electron backscattered diffraction (EBSD) was used for the measurement.

To measure the tensile properties, test pieces with 50-mm parallel length were machined according to ASTM E-8, and the tensile tests were performed with a crosshead speed of 2 mm/min.

To measure the damping capacity, specimens with the dimensions of 120 mm × 10 mm × 2 mm were manufactured by electric discharge machining from a cold-rolled plate with different cold rolling degrees after the solution treatment. Afterward, an internal friction measurement machine (IFT-1500, Ulvac, Chigasaki, Japan) was used to evaluate the logarithmic decay rate. The logarithmic decay rate (*δ* = 1/*n* ln*A*_0_/*A_n_*, where *n* is the wavenumber, *A*_0_ is the amplitude of the first wave, and *A_n_* is the amplitude of the nth wave) was obtained by measuring the wavenumber [15].

## 3. Results and Discussion

Figure 1 represents the optical microstructure of the as-received Fe–20Mn–12Cr–3Ni–3Si damping alloy. Certain volume fractions of martensite in the austenite structures, including some twin crystals, were detected, which mainly consisted of the austenite phase due to the solution treatment at 1050 °C. 

Figure 2 shows the OM and SEM images of the 29% cold-rolled specimen. A larger volume fraction of martensite in this specimen was identified than that in the as-received specimen, since some of the austenite phases deformed the martensite phase through cold rolling. The microstructural observation confirmed that martensitic phases were generated, causing surface relief with a specific orientation, which is known as a phenomenon relating to the martensitic transformation [16], and partially intersecting with each other.

Figure 3 illustrates a TEM bright-field image and the selected area diffraction pattern of the austenite phase in a basal substrate structure and the ε-martensite phase obtained by 49% cold rolling after solid solution treatment. The ε-martensite with the HCP crystal structure was transformed with the orientation relationship between the austenite (γ) [110] and ε-martensite [10$\bar{1}$0]. With a high cold rolling degree in this alloy, it was discovered that several ε-martensite phases with different orientations formed in the transformation from austenite. Regarding the ε-martensitic transformation induced by deformation, in the early stage of deformation, dislocations remaining in the matrix after heat treatment tended to be active as dislocation sources in the main slip system, which favored deformation. Several perfect and extended dislocations piled up on a single slip surface. As the deformation progressed, extended dislocations in the main slip system were generated to fill the gaps in the slip plane where extended dislocations had already been introduced. As the deformation degree proceeded, the partial dislocations of the same component migrated in layers until they were sufficient to form the HCP structure and generate the deformation-induced ε-martensite.

To understand the transformation behavior of α′-martensite, TEM observation was examined using the specimen deformed by 49% cold rolling (Figure 3c,d). The formation of ε martensitic plates and the intersection with each plate containing α′ martensitic transformation were identified with the indexing of [110]. Due to the TEM observation and pattern indexing, α′-martensite was mainly transformed in the intersection of ε martensitic plates in this alloy. This is in good agreement with previous results [17,18,19,20] that reported that one of the ε martensitic plates contains small regions of austenite, whereas two ε-martensite plates intersect to form α′-martensite.

Note that there is controversy over the nature of the α′ martensitic transformation. Some studies [21,22,23] assert the direct transformation from austenite (γ), whereas others [17,19,20] have contended the sequential γ → ε → α′ martensitic transformation, which is consistent with this study. It will be necessary to investigate direct experimental pieces of evidence for α′ martensitic transformation.

To investigate the volume fraction variation for each phase as a function of the deformed ratio, the volume fractions of each phase were evaluated by the calculation [14] of X-ray diffraction as shown in Figure 4. As the deformation degree increased, it is not surprising that the fractions of αʼ-martensite and ε-martensite gradually increased, whereas the fraction of the austenite phase decreased. These variations can be explained by the fact that the deformation-induced α′ and ε martensitic transformations from certain austenite phases occur.

Besides, the volume fraction variation for each phase depicts that α′-martensite transformed after a 15% deformation degree and then slightly increased, whereas ε-martensite formed at the early deformation stage and gradually increased. Note that the volume fraction of α′-martensite was < 6%, whereas that of ε-martensite indicated 60% when the deformation degree approached 68%. This difference in volume fraction between α′ and ε-martensite may be caused by the high austenite stability as a result of adding alloying elements, such as Mn and Ni, to this alloy. Mn and Ni are effective elements that lower the martensite start temperature (Ms) (Table 2) and stabilize the ε-martensite [24]. Moreover, the difference in volume fraction between the α′-martensite and ε-martensite can also be explained by reducing the stacking fault energy ($SFE,\;\gamma_{sf})\;$through adding alloying elements [25,26].

Depending on the alloying element herein, the stacking fault energy ($\gamma_{sf}$) was calculated to be 11.0 mJ/m^2^ using the following formula equation [32],
(1)$$\gamma_{sf}=28.87+1.64\;(\times \;\mathrm{wt}.\%\;\mathrm{Ni})-1.1\;(\times \;\mathrm{wt}.\%\;\mathrm{Cr})+0.21\;(\times \;\mathrm{wt}.\%\;\mathrm{Mn})-4.45\;(\times \;\mathrm{wt}.\%\;\mathrm{Si})$$

Even though Mn and Ni increase the SFE, Cr and Si are more effective in reducing the SFE, as given in the above formula and other studies [32,33,34]. It has also been reported that Ni is effective in retarding α′-martensite formation [35].

Generally, a stacking fault energy value of <20 mJ/m^2^ leads to TRIP, followed by the formation of deformation-induced HCP ε-martensite [22] since the stacking faults and dislocations are crucial as nucleating seeds of ε-martensite [19].

Figure 5 represents the tensile strength and elongation as a function of the cold rolling degree of this alloy. The tensile strength linearly increased as the ratio of cold rolling increased. Besides, it is interesting to note that 0% elongation occurred when the cold rolling degree was ~68%. The increase in strength and decrease in elongation occurs because the volume fraction of martensite, which is associated with high strength, increases, whereas that of austenite decreases as the cold rolling degree increases. This trend shows excellent agreement with the behavior in deformation-induced martensitic transformation alloys [36,37].

To verify the cold rolling effect on the damping capacity, the internal friction values of alloys with varying cold rolling degrees were investigated (Figure 6). The internal friction values increased until they approached culmination at a 30% cold rolling degree, and then decreased. The relationship between ε-martensite and internal friction reveals a similar trend, where the internal friction increased until the volume fraction of ε-martensite approached 60%, and then decreased. This trend is dissimilar to other assertions [38], demonstrating that the damping capacity (internal friction) increases as the volume fraction of ε-martensite increases since the damping capacity can be closely influenced by the ε-martensite content. This abnormal trend shown by the damping capacity in this study can be explained by the fact that dislocations generated during the cold rolling degrees of >30% obstruct the movement of ε-martensite-related damping sources, such as interface boundaries between austenite and ε-martensite, stacking fault boundaries in ε-martensite, and the variant boundary of martensitic plates in alloys [39].

Although the ε-martensite influences the damping capacity, controversy exists over the effect of the α′-martensite on the damping capacity. Some studies reported that the strain-induced α′-martensite influences the damping properties in austenitic stainless steel [40,41], whereas the damping capacity in high-Mn alloy seems not to be attributed to the α′-martensite [38]. Herein, it is hard to clarify the effect of α′-martensite on the damping capacity because the volume fraction of α′-martensite is too low (<6%) to evaluate the effect.

Figure 7 illustrates the relationship between the internal friction and the tensile strength/elongation in this alloy. As the tensile strength and elongation increased, the damping capacity increased to specific values and then decreased. This result is inconsistent with the general tendency that the tensile strength is inversely proportional to the internal friction [42]. It is well-known that the damping capacity tends to decrease since it is difficult to dissipate vibration energy into thermal energy in alloys with high strength [5,8]. However, in the area of high tensile strength, the decreased internal friction can be explained by the above description. In contrast, in the area of low tensile strength, it is inconsistent with the general tendency, depicting that as the tensile strength increases, the internal friction increases. It can be suggested that the sharp increase in the ε-martensite volume fraction increases the damping source that is associated with stacking faults in ε-martensite and the interface between austenite and ε-martensite. Accordingly, it can be concluded that the damping capacity increases to specific values and then decreases as the tensile strength and elongation increase.

## 4. Conclusions

Herein, the tensile strength and damping capacity of Fe-20Mn-12Cr-3Ni-3Si alloy cold-rolled at different degrees were investigated. The following conclusions can be drawn.

In this alloy, martensitic transformations occurred by cold rolling, including surface relief with a specific orientation. As the cold rolling degree increased, the volume fractions of ε-martensite and austenite (*γ*) increased and decreased, respectively, in this alloy. α′-martensite started gradually forming at the cold rolling degree with 15% and increased to 6% at the maximum cold rolling degree in this alloy. The difference in volume fraction may be caused by the high austenite stability as a result of adding alloying elements, such as Mn and Ni, which reduces the stacking fault energy.The tensile strength and elongation increase and decrease as the volume fraction of martensite increases due to the increase in the cold rolling degree. The damping capacity increases to a certain degree and then decreases since dislocations generated during the cold rolling degrees > 30% obstruct the movement of ε-related damping sources in this alloy.In the high-strength area, the damping capacity tends to decrease because it is difficult to dissipate vibration energy into thermal energy. On the other hand, in the low-strength area, the increased volume fraction of ε-martensite is attributed to the increase in the damping source associated with stacking faults in ε-martensite and the interface between austenite and ε martensite. Accordingly, the damping capacity increases until it culminates and decreases as the tensile strength and elongation increase.

## Figures and Tables

**Figure 1 materials-14-05975-f001:**
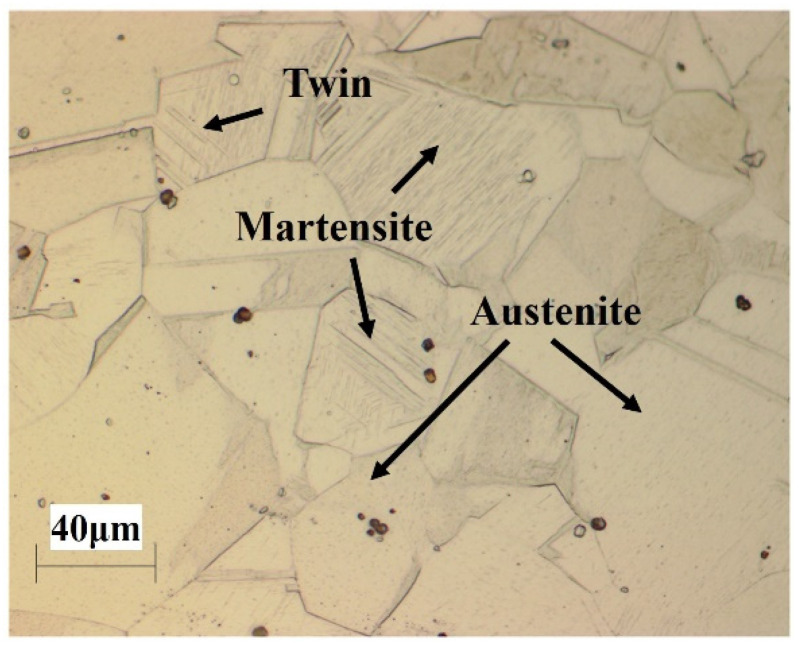
OM image of solution-treated specimen.

**Figure 2 materials-14-05975-f002:**
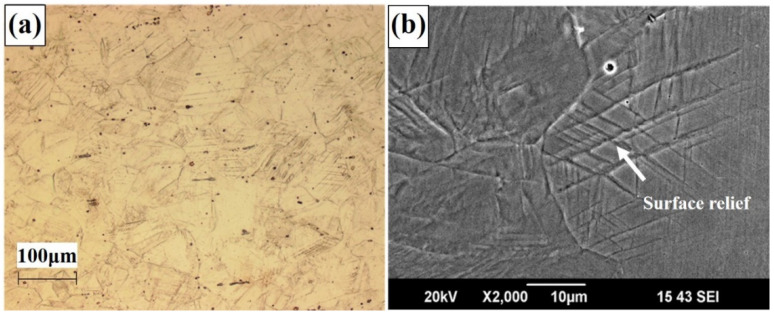
(**a**) OM and (**b**) SEM images of 29% cold-rolled specimen.

**Figure 3 materials-14-05975-f003:**
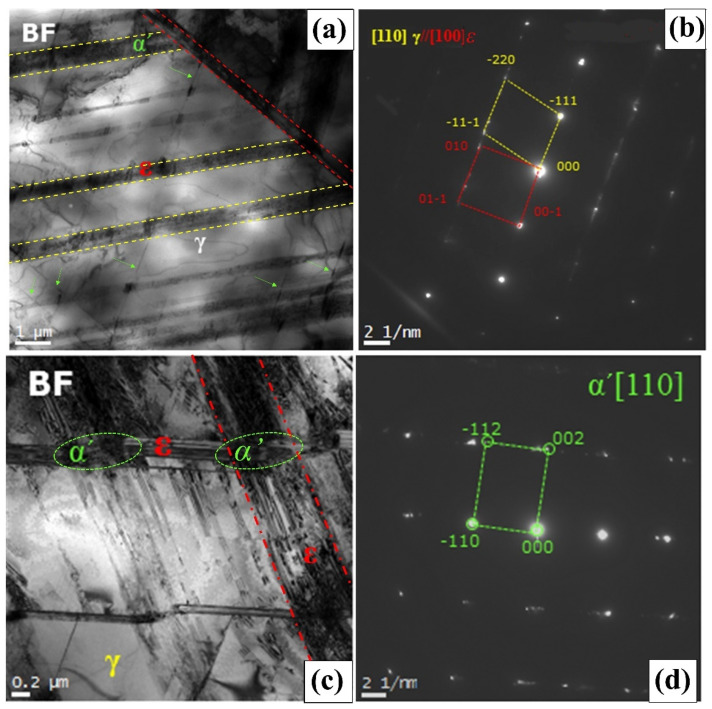
TEM images of ε-martensite and α′-martensite obtained in 49% cold-rolled Fe–20Mn–12Cr–3Ni–3Si alloy with (**a**) bright-field image of ε-martensite, (**b**) SADP and its index, (**c**) bright-field image of α′-martensite, and (**d**) SADP and its index.

**Figure 4 materials-14-05975-f004:**
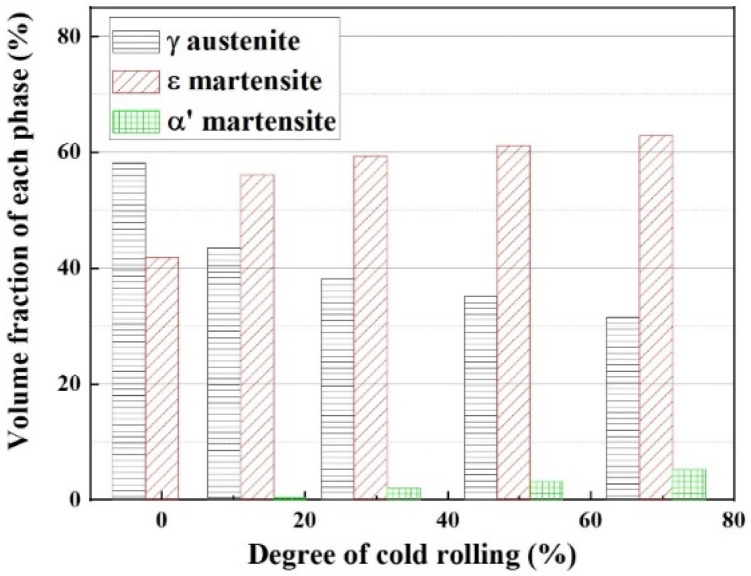
Volume fraction of each phase as a function of cold rolling degree in Fe–20Mn–12Cr–3Ni–3Si alloy.

**Figure 5 materials-14-05975-f005:**
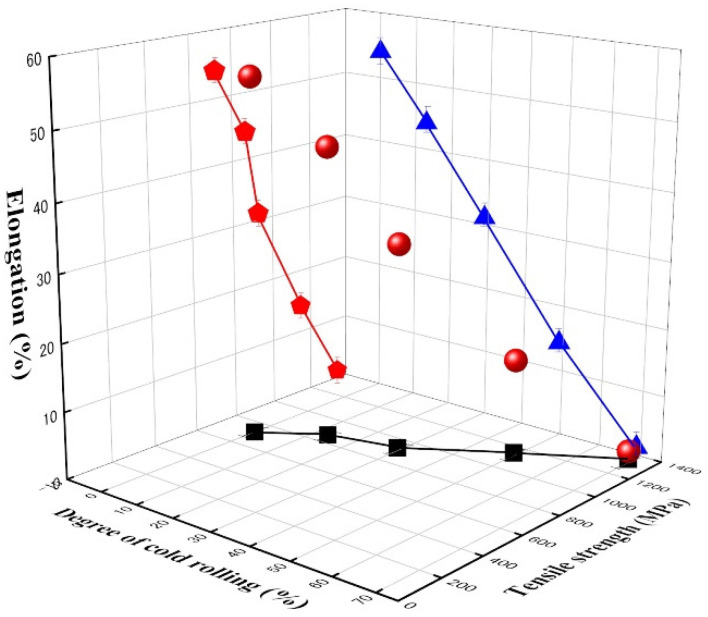
Tensile strength and elongation as a function of cold rolling degree.

**Figure 6 materials-14-05975-f006:**
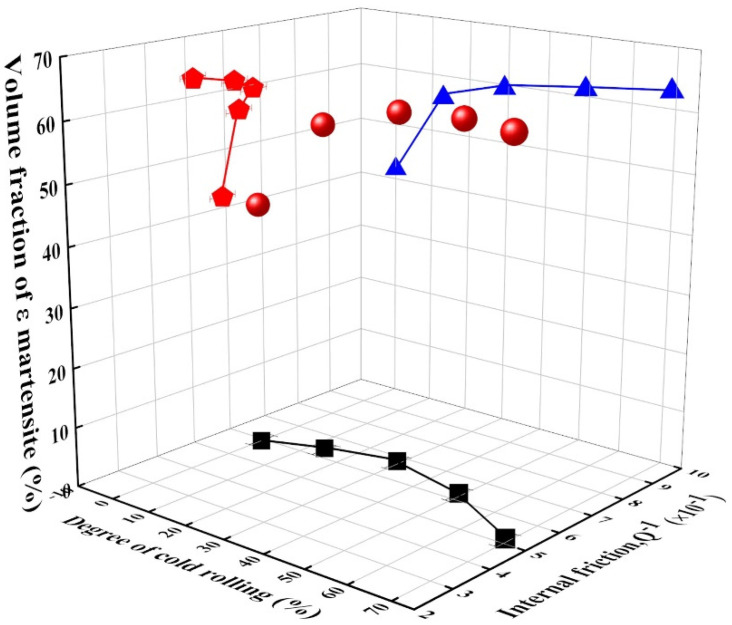
Volume fraction of ε-martensite and internal friction as a function of cold rolling degree.

**Figure 7 materials-14-05975-f007:**
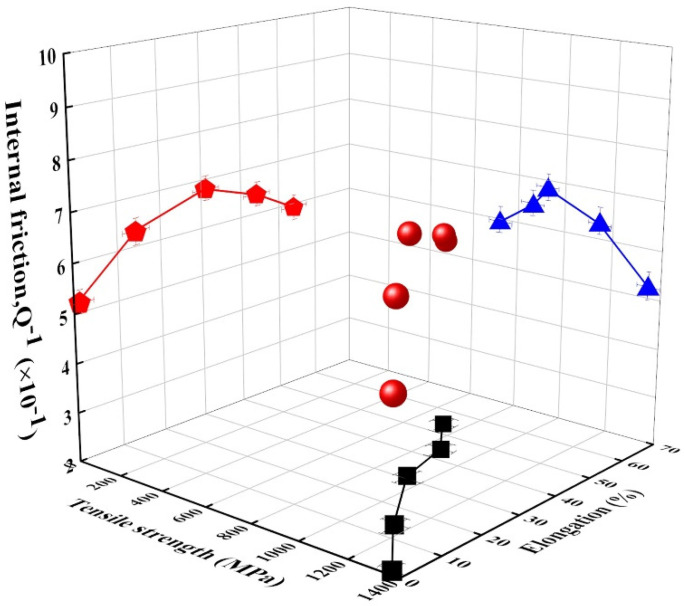
Internal friction according to tensile strength and elongation variations in cold-rolled alloy.

**Table 1 materials-14-05975-t001:** Chemical compositions.

Compositions	C	N	P	S	Mn	Cr	Ni	Si	Fe
wt.%	0.01	0.02	0.001	0.008	20.3	12.08	3.2	3.15	bal.

**Table 2 materials-14-05975-t002:** Ms and As of Fe–Mn alloys.

Composition.	Ms (K)	As (K)	Method	Reference
Fe-20.4Mn-12.7Cr	340	416	Measured	[27]
Fe-20.0Mn-3.0Si	420	490	Calculated	[28]
Fe-17.5Mn-1.9Si	436	490	Measured	[28]
Fe-19.5Mn-2.0Si	420	480	Measured	[28]
Fe-24.2Mn-1.9Si	399	465	Measured	[28]
Fe-25.9Mn-1.8Si	358	445	Measured	[29]
Fe-20.8Mn	387		Calculated	[30]
Fe-20.0Mn-1.1Si	408	469	Measured	[28]
Fe-19.5Mn-2.0Si	420	480	Measured	[28]
Fe-18.9Mn-3.2Si	430	490	Measured	[29]
Fe-17.3Mn-0.9Ni	395	525	Measured	[31]
Fe-17.5Mn-1.4Ni	369	514	Measured	[31]
Fe-17.5Mn-2.0Ni	349	501	Measured	[31]

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
