# Peer review of "Tensile Properties and Damping Capacity of Cold-Rolled Fe-20Mn-12Cr-3Ni-3Si Damping Alloy"

_materials, 2021, doi:10.3390/ma14205975_

Round 1
Reviewer 1 Report
Comments and Suggestions for Authors
The paper concerns the investigation of the development of steel materials damping alloy (Fe–20Mn–12Cr–3Ni–3Si), evaluated by mechanical properties, such as strength and damping capacity. And, the authors focused about the mechanical properties in this alloy to determine the relationship between the tensile properties and damping capacity.
The scientific results are many, but the work needs a revision to better clarify the interpretation of the data and to better present the experimental results. The author should read the manuscript and clarify the command clearly. The authors have to care about typos. There are careless mistakes in some places.
- In the abstract, L22: …the damping capacity, the damping capacity….Please rewrite the sentence.
- The following papers should be examined and referred; - Heusler-type magnetic shape memory alloys: a review. Inter J Adv Manuf Techno (2019). Doi: 10.1007/s00170-019-03534-3.- Influence of Chemical Composition on martensitic transformation of MnNiIn Shape Memory Alloys. Journal of Thermal Analysis and Calorimetry (2015) DOI 10.1007/s10973-015-4716-8. - Effects of Co Additions on the Martensitic Transformation and Magnetic properties of Ni-Mn-Sn Shape Memory Alloys. J Super and Nov Mag (2015). DOI: 10.1007/s10948-015-3100-z.
- Please add the XRD patterns of the studied alloy (Fe–20Mn–12Cr–3Ni–3Si).
- Please complete the sentence in L149: As the deformation degree increases, it is not surprising that the…
Reviewer 2 Report
Dear Editor: I would like to express my deep thanks for inviting me to review the manuscript ID: materials-1398751
Title: Tensile properties and damping capacity of cold-rolled Fe- 20Mn-12Cr-3Ni-3Si damping alloy
Authors: Jae-Hwan Kim1, Jong-min Jung and Hyunbo Shim
Comments:
Abstract:
Please rewrite the abstract according to your results.
Introduction:
There is no sufficient information. Please rewrite the introduction section.
Please read these damping capacity articles and cited
- Zhang, R.J. Perez, E.J. Lavernia Documentation of damping capacity of metallic, ceramic and metal-matrix composite materials, J. Mater. Sci.,28 (1993), 2395-2404
- AK Gain, L Zhang, Effect of Ag nanoparticles on microstructure, damping property and hardness of low melting point eutectic tin–bismuth solder, J. Mater. Sci.: Mater. Electron. 28 (20), (2017) 15718-15730.
- H. Chang, S.K. Wu, Low-frequency damping properties of eutectic Sn–Bi and In–Sn solders, Scripta Mater., 64 (2011), 757-760
Clearly explain the objectives and novelty
Results and discussion:
- SEM image in Figure 2 (b) is not clear. Instead of SEM image provide EBSD images with phase that will be helpful for reader.
- Please combine TEM images Figure 3 & 4 and explain in detail the formation mechanism of martensite.
- Plead add STD in Figure 5.
- Please modify the Figure 7 and Figure 8 3D image to line graph with error bar
Conclusion part:
Please concise the conclusion parts.
RECOMMENDATION
After reviewing the enclosed manuscript for “Materials”, the present manuscript contains some kinds of scientific analysis but it is mandatory required to modify according to the preceding remarks. So, the manuscript can be publication after major revision.
Reviewer 3 Report
Unfortunately, the quality of the work cannot be assessed, since Figures 6-8 are absolutely unclear and do not contain comments. The authors sent a completely unfinished article to the editorial office. To evaluate the work, the authors should carefully double-check it.
Author Response
It was revised based on the reviewer's comments. In addition, the authors tried to concise the contents in a manuscript.
Please see the attachment.

Round 2
Reviewer 2 Report
Dear Editor: I would like to express my deep thanks for inviting me to review the manuscript ID: materials-1398751
Title: Tensile properties and damping capacity of cold-rolled Fe- 20Mn-12Cr-3Ni-3Si damping alloy
Authors: Jae-Hwan Kim1, Jong-min Jung and Hyunbo Shim
Comments:
Abstract:
Please rewrite the abstract according to your results.
Introduction:
There is no sufficient information. Please rewrite the introduction section.
Please read these damping capacity articles and cited
- Zhang, R.J. Perez, E.J. Lavernia Documentation of damping capacity of metallic, ceramic and metal-matrix composite materials, J. Mater. Sci.,28 (1993), 2395-2404
- AK Gain, L Zhang, Effect of Ag nanoparticles on microstructure, damping property and hardness of low melting point eutectic tin–bismuth solder, J. Mater. Sci.: Mater. Electron. 28 (20), (2017) 15718-15730.
- H. Chang, S.K. Wu, Low-frequency damping properties of eutectic Sn–Bi and In–Sn solders, Scripta Mater., 64 (2011), 757-760
Clearly explain the objectives and novelty
Results and discussion:
- SEM image in Figure 2 (b) is not clear. Instead of SEM image provide EBSD images with phase that will be helpful for reader.
- Please combine TEM images Figure 3 & 4 and explain in detail the formation mechanism of martensite.
- Plead add STD in Figure 5.
- Please modify the Figure 7 and Figure 8 3D image to line graph with error bar
Conclusion part:
Please concise the conclusion parts.
RECOMMENDATION
After reviewing the enclosed manuscript for “Materials”, the present manuscript contains some kinds of scientific analysis but it is mandatory required to modify according to the preceding remarks. So, the manuscript can be publication after major revision.
Reviewer 3 Report
Unfortunately the authors did not make any significant corrections to the sections proposed by me and other reviewers, the article is still not possible to evaluate
Author Response
Because the authors could not receive any detailed comments from reviewer 3, the authors revised some parts commented by reviewer 2 and submitted the revised version. There is no response to reviewer 3.